# The Role of Resveratrol Administration in Human Obesity

**DOI:** 10.3390/ijms22094362

**Published:** 2021-04-22

**Authors:** Laura M. Mongioì, Sandro La Vignera, Rossella Cannarella, Laura Cimino, Michele Compagnone, Rosita A. Condorelli, Aldo E. Calogero

**Affiliations:** Department of Clinical and Experimental Medicine, University of Catania, Policlinico “G. Rodolico”, Via S. Sofia 78, 95123 Catania, Italy; lauramongioi@hotmail.it (L.M.M.); sandrolavignera@unict.it (S.L.V.); rossella.cannarella@phd.unict.it (R.C.); lauracimino@hotmail.it (L.C.); michele.compagnone22@tiscali.it (M.C.); rosita.condorelli@unict.it (R.A.C.)

**Keywords:** resveratrol, obesity, clinical trials

## Abstract

Obesity is a widespread disease that is associated with numerous and serious comorbidities. These include metabolic syndrome, diabetes mellitus, cardiovascular-cerebrovascular disease, hypertension, obstructive sleep apnea syndrome, cancer, and sexual and hormonal disorders. The treatment of obesity has therefore become a goal of great clinical and social relevance. Among the therapeutic strategies against obesity, resveratrol has aroused great interest. This polyphenol has anticancer and antioxidant properties and cytoprotective and anti-inflammatory effects. Other favorable effects attributed to resveratrol are anti-lipid, anti-aging, anti-bacterial, anti-viral, and neuroprotective actions. Administration of resveratrol appears to improve the metabolic profile in obese and/or insulin-resistant patients. This article aims to review the main results of clinical studies evaluating the effects of administering resveratrol alone in overweight/obese patients.

## 1. Introduction

In recent years, weight gain has become a troubling reality as the number of overweight/obese people around the world has reached epidemic proportions [1]. The prevalence of obesity in European countries is around 20%, with no significant differences between men and women. Available data show that weight gain is particularly prevalent in Western countries mainly due to the combination of high energy food intake and low levels of physical activity [2,3]. Obesity is an important risk factor for chronic diseases, such as metabolic syndrome, diabetes mellitus, cardiovascular/cerebrovascular diseases, hypertension, obstructive sleep apnea syndrome, oncological diseases, and sexual and hormonal disorders [1]. Weight gain is also linked to accelerated skeletal growth and pubertal development and is associated with the aging process. In this regard, the release of inflammatory cytokines appears to be related to the shorter lifespan reported in obese patients [3].

Lifestyle changes, mainly in the form of dietary intervention and physical activity, are undoubtedly necessary. However, to date, preventive measures often fail to reduce the onset of obesity and the available therapeutic strategies are not as effective. For these reasons, interest in new approaches, such as the use of nutraceuticals, is growing. Among these, resveratrol has aroused much interest due to its multiple beneficial effects reported in humans [4].

Resveratrol (trans-3,5,4’-trihydroxystilbene) is a polyphenolic compound that, in 1939, was isolated from the roots of white hellebore [5]. Later, it was found in peanuts, grapes, blueberries, rhubarb, and wine [6]. In the 90s, resveratrol was considered the origin of the so-called “French paradox”. This is the phenomenon whereby in the French population, despite a diet rich in fat, there was a low prevalence of obesity and cardiovascular diseases attributed to the moderate consumption of red wine, rich in this polyphenol [5].

The scientific interest in resveratrol dates back to 1997 when studies on animal models demonstrated its anticancer properties [7]. Resveratrol also has cytoprotective effects and antioxidant properties, and modulates anti- and pro-apoptotic mediators, protecting cells from DNA damage. This polyphenol also has anti-inflammatory effects. In fact, it inhibits cyclooxygenase-2 (COX-2), nuclear factor-kB (NF-kB), and other molecules involved in inflammatory processes [6,8]. Other favorable effects attributed to resveratrol are anti-lipid, anti-aging, anti-bacterial, anti-viral, and neuroprotective action. Finally, supplementation of resveratrol appears to improve the metabolic profile in obese and/or insulin-resistant patients [3,4,5].

Although many studies have investigated the beneficial effects of resveratrol on experimental animals, human clinical trials undertaken to evaluate the effect of this polyphenol on weight gain are few and often with controversial results. A 2016 systematic review, including studies published between 1990 and November 2015, concluded that the evidence to support the beneficial role of resveratrol in obesity is insufficient in terms of body weight and body mass index (BMI) decrease. However, the review included only nine studies for a total of 208 patients [9]. More recently, a systematic review with meta-analysis investigated the effects of resveratrol on weight loss, concluding that its administration significantly reduces body weight, BMI, waist circumference, and fat mass [10]. In contrast, a subsequent systematic review with meta-analysis found no anti-obesity effects of resveratrol [11]. However, this review included studies on patients with comorbidities and some of the studies included did not have obesity among the inclusion criteria of the patients enrolled. Moreover, other studies evaluated the effect of the co-administration of resveratrol with other substances [11]. Specifically, among the 19 studies included in the meta-analysis, two evaluated the effect of resveratrol in association with other compounds, five did not consider BMI among the inclusion criteria, two did not evaluate the BMI post-treatment, and two enrolled non-obese patients.

Therefore, given that obesity is a disease with complex and multifaceted etiopathogenesis, this review evaluated the effects of administering resveratrol alone in patients with obesity. We also reviewed the mechanisms of action of this polyphenol in weight gain.

## 2. Resveratrol and Obesity: Possible Mechanisms of Action

In recent years, many authors have investigated the effects of resveratrol in obesity and metabolic syndrome, reporting an improvement in glucose homeostasis and the cardiovascular risk associated with obesity. However, the exact mechanisms by which resveratrol carries out its beneficial effect have not been fully understood. Resveratrol may affect the activity of various intracellular targets such as adenosine monophosphate-activated protein kinase (AMPK), the deacetylating enzyme sirtuin-1 (SIRT-1), and the peroxisome proliferator-activated receptor γ coactivator-1α (PGC-1α), all altered in the metabolic abnormalities present in patients with obesity [12,13]. In particular, resveratrol can influence gene expression by inducing changes that mimic those seen under calorie restriction conditions [12].

Experiments in animal models have shown that resveratrol significantly reduces body weight [14]. In these models, it has effects similar to those induced by a calorie restriction regimen. It also improves mitochondrial function, which is compromised in numerous metabolic diseases, and cellular sensitivity to insulin in mice fed a high-energy diet [3].

In the context of obesity, resveratrol appears to be responsible for modulating mitochondrial activity for which activation of SIRT-1 plays a key role. In mammals, SIRT-1 acts as a regulator of healthspan, protecting against aging and stressful metabolic conditions such as impaired glucose homeostasis, obesity, and cancer [13]. SIRT-1 is a human deacetylase that negatively regulates the p53 tumor suppressor gene, thus it promotes cell survival [14]. The levels of SIRT-1 are higher in the muscular tissue in the presence of caloric restriction (calorie intake lower of about 20–40%) [15]. The direct activation of SIRT-1 by resveratrol is controversial since in-vitro experiments have shown that the interaction between resveratrol and the artificial fluorescent substrate used could generate artifacts [15]. Thus, other possible pathways were searched to evaluate whether resveratrol could activate SIRT-1 indirectly.

Among these, numerous pieces of evidence suggest that resveratrol activates AMPK [13,16]. AMPK can be activated by an increased ADP/ATP ratio, as occurs in states of energy deficiency by increasing intracellular NAD^+^, a substrate of SIRT-1 [16]. Another possibility is that resveratrol may stimulate AMPK by activating two proteins required for AMPK phosphorylation: LKB1 and calcium/calmodulin-dependent kinase kinase β (CamKKβ). The former activates AMPK in case of ATP depletion and the consequent intracellular increase of Ca^2+^ stimulates CamKKβ, which in turn can activate AMPK [16]. However, resveratrol-induced AMPK phosphorylation appears to be high-dose dependent [12]. Indeed, in a SIRT-1 knock-out mouse model, resveratrol is unable to induce AMPK phosphorylation at a low dose. In contrast, AMPK phosphorylation was observed at 10 times higher dose. These results, therefore, indicate that resveratrol can phosphorylate AMPK in a dose-dependent manner and independently of SIRT-1 [17]. This is an important finding since the phosphorylation state of AMPK represents a signal for adipogenesis and its phosphorylation hinders this metabolic pathway [12].

There is general agreement on the inhibitory effect that resveratrol has on adipogenesis, as numerous studies have examined this aspect in-vitro in animal models and most of them have shown concordant results. Lasa and colleagues [18] incubated pre-adipocyte cells with increasing concentrations of resveratrol (1, 10, and 25 µM) for 24 h, reporting a decrease in triacylglycerol content and a lower expression of acetyl-CoA carboxylase (ACC). The first is an enzyme with a key role in the synthesis of long-chain saturated fatty acids. In the adipose tissue, lipoprotein lipase, which is activated by the peroxisome proliferator-activated receptor-gamma (PPARγ), catalyzed the hydroxylation of triacylglycerol in free fatty acids. Hence, these results suggest the presence of an inhibitory effect of resveratrol on lipoprotein lipase [18]. Consistent with this, another study reported significant downregulation of lipoprotein lipase mRNA in peripheral blood mononuclear cells (PBMC) of pigs after daily administration of resveratrol at a dose of 0.11 mg/kg of body weight for 12 months [19]. Other animal studies, mainly conducted in rats, have shown a reduction of abdominal fat, lipoprotein lipase, and ACC activities after treatment with resveratrol [20,21,22]. Specifically, Rivera and colleagues evaluated the effects of resveratrol on de-novo lipogenesis in obese and lean 13-week-old Zucker rats. They received a dose of 10 mg/kg of body weight for 8 weeks. No changes in food intake were observed in the treated mice. Obese rats were found to have lower abdominal fat content than pre-treatment values, while no change was found in lean ones. Also, an increase in phosphorylated ACC protein expression was found. Since it is an inactive form of ACC, the authors suggested that resveratrol decreased ACC activity and thus de-novo lipogenesis [20].

SIRT-1 can also increase the activity of PGC-1α, which in turn induces gluconeogenesis and glycolysis in the liver by increasing the transcription of glycogenetic genes [23]. Some evidence indicates that resveratrol stimulates mitochondrial activity through SIRT-1-mediated activation of PGC-1α, which contributes to fiber-type switching and adaptive thermogenesis in muscles and brown adipose tissue, respectively [23]. Mitochondrial activity is triggered by physical activity and dietary restrictions, two cornerstones of the medical approach to obesity. Chronic physical activity induces a reduction in lipid storage, mobilization of fatty acids, reduction of inflammatory adipokines, as well as an improvement in mitochondrial function in white adipose tissue [24]. Furthermore, a low-calorie diet could activate mitochondrial oxidation [25]. Therefore, resveratrol appears to activate the same metabolic pathways commonly stimulated by available and validated therapies for obesity. This further increases the likelihood this compound could be an effective weapon for the management of obesity.

Resveratrol can also indirectly inhibit adipocyte differentiation by modulating their gene expression [26]. Being an acetylase and therefore acting epigenetically, SIRT-1 can influence gene expression by acetylating their promoter region. Resveratrol is in fact able to induce the expression of genes that promote oxidative phosphorylation and mitochondrial biogenesis as a result of SIRT-1 [23]. In the process of differentiation of adipocytes from stem cells, certain proteins such as CCAAT-enhancer-binding protein (C/EBPα), sterol regulatory element-binding protein 1c (SREBP-1c), and PPARγ play a fundamental role in promoting changes from fibroblastic form to spherical cell [27]. Some authors found that after incubation of pre-adipocytes with resveratrol at concentrations >10 μM, gene and protein expression of C/EBPα, SREBP-1c, and PPARγ were down-regulated, thereby reducing adipogenesis. Concentrations between 25–50 μM have been shown to significantly reduce the lipid accumulation in pre-adipocytes after 6 days of incubation, due to both decreased adipogenesis and reduced cell viability [27]. These findings were confirmed by other authors, who used different dosages (10–40 μM) and incubation times (2, 4, and 6 days). The greatest amount of inhibition of pre-adipocyte differentiation was 40% and was achieved at a dose of 40 μM for 6 days [28]. It has been hypothesized that the inhibition of adipogenesis may also be mediated by SIRT-1 directly or indirectly by increasing the expression of the Forkhead box protein O1 (FoxO1) [28]. Some studies have also reported an increase in adipocyte apoptosis via the SIRT-1/AMPK axis and a lipolytic effect of resveratrol, although the latter effect is still debated [27].

Interestingly, it has been suggested that the effect of resveratrol on adipocyte differentiation may be somehow mediated by the glucocorticoid receptor. Hu and colleagues studied the effect of incubating with 1, 10, 50, and 100 μM of resveratrol, for 7 days, using a culture medium containing cortisone, a glucocorticoid receptor agonist. The author reported a dose-dependent stimulation of adipocytes differentiation at concentrations of 1 and 10 μM. Conversely, an inhibitory effect was found at concentrations of 50 and 100 μM, as confirmed by the reduction in the expression of C/EBPα, FABP4, and PPARγ. Based on their findings, the authors hypothesized that the effect of resveratrol on adipocytes may be mediated by activation of the glucocorticoid receptor [29].

Resveratrol has also been shown to promote the differentiation of white adipocytes into brown ones [23]. However, the effects of resveratrol on adipogenesis and adipocyte differentiation could hide pitfalls. Although the inhibition of adipogenesis may be considered an interesting strategy for obesity prevention, some evidence highlights that the disruption of adipocyte differentiation and expansion could lead to ectopic fat accumulation and metabolic alterations, such as insulin resistance and type 2 diabetes mellitus [30]. Thus, it has been hypothesized that adipose tissue expansion and increased adipogenesis may be associated with beneficial effects on adipocyte endocrine function and, consequently, on the metabolic status [30]. Interestingly, an agonist of PPARγ, an ethanolic extract of Artemisia scoparia (SCO), showed both in-vitro and in-vivo beneficial effects on insulin sensitivity and adipocyte function [30]. Impaired adipocyte differentiation, moreover, is also associated with inflammatory processes in the adipose tissue that are well-known pathogenetic aspects of obesity [31]. Furthermore, SCO seems to interfere with the inflammatory pathway, thus reducing metabolic dysfunction, and has a favorable effect on pancreatic β-cells [31]. Therefore, the investigation on the possible combined effects of resveratrol and SCO on adipogenesis and adipose tissue metabolism could represent an interesting challenge for the future.

As for other mechanisms of action of resveratrol, experimental evidence has shown that this polyphenol can inhibit the activity of phosphodiesterases (PDE) and in particular of the PDE4 isoform, thus increasing the intracellular levels of cyclic adenosine monophosphate (cAMP) [17]. PDE regulate the signaling of pancreatic β-cell within which they modulate the Ca^2+^ levels and, consequently, the secretion of insulin [16]. The increase of cAMP levels activates protein kinase A (PKA) and exchange protein activated by cAMP (EPAC), inducing the activation of AMPK e consequently of SIRT-1 [16]. 

In summary, these findings strongly suggest a role for resveratrol in inhibiting adipogenesis and glycolysis, and in stimulating gluconeogenesis and mitochondrial activity (Figure 1). Similar metabolic pathways are also triggered by physical activity and calorie restriction with diet. Despite this evidence, many aspects need to be clarified and therefore further studies are needed to better understand the mechanisms of action of resveratrol.

## 3. Bioavailability and Pharmacokinetics

As reported for other polyphenols, resveratrol has a low bioavailability [8]. Although oral absorption is approximately 75%, it is actually not easy to identify unmetabolized resveratrol in human plasma [32]. During supplementation, there is a dose-dependent response between the amount of resveratrol administered and its Cmax. Studies examining the pharmacokinetics of this polyphenol have led to results that are difficult to interpret due to the different dosages administered [12]. Multiple daily doses increase the Cmax of resveratrol and it appears that this polyphenol and its metabolites accumulate in human cells in a dose-dependent and tissue-specific manner [8]. However, there are inter-individual variations in its bioavailability and, to date, the exact role of its metabolites is unclear [8]. In in-vitro experiments, resveratrol concentrations range from 5–100 µM, although the concentration of 50 µM is the most commonly used [12].

## 4. Obesity and Resveratrol: Clinical Trials in Humans

Despite numerous studies conducted on experimental models in animals, only a few trials have evaluated the effects of resveratrol administered alone and not in combination with other nutraceuticals on obesity in humans.

Timmers and colleagues were among the first researchers to demonstrate that resveratrol administration causes metabolic changes similar to those seen in calorie restriction regimens [13]. In their placebo-controlled, double-blind cross-over study, 11 otherwise healthy obese patients (52.5 ± 2.1 years) were treated with placebo and subsequently, after a 4-week wash-out period, with 150 mg of resveratrol once daily for 30 days. Each patient underwent measurements of serum levels of dihydro-resveratrol, total (conjugated and unconjugated) and free resveratrol to assess the patients’ compliance to the study protocol and the adsorption of resveratrol. At the end of treatment, the administration of resveratrol significantly reduced blood glucose, insulin concentrations, the homeostasis model assessment (HOMA) index, triglycerides, sleeping and resting metabolic rate, and blood pressure. Also, liver function improved. No effect was found on body weight. Resveratrol up-regulated the expression of genes linked to mitochondrial oxidative phosphorylation, while it down-regulated those involved in inflammation. After muscle biopsy, the authors also found increased AMPK phosphorylation and SIRT-1 and PGC-1α expression. No adverse reactions were reported [13].

In 2011, in an uncontrolled, open-label pilot study, 10 overweight/obese adults (72 ± 3 years) with moderate insulin-resistance, diagnosed after a 75 g oral glucose tolerance test, were randomly assigned to take resveratrol at the doses of 1, 1.5, or 2 g once a day for 4 weeks [33]. Resveratrol had no significant effect on fasting blood glucose levels, insulin secretion, HOMA index, blood pressure, fasting lipid profile, c-reactive protein, adiponectin, and body weight, but significantly decreased postprandial glucose peak. Endothelial function, assessed by reactive hyperemia of peripheral arterial tonometry, improved but not significantly. Regarding adverse reactions, one patient reported only mild diarrhea [33].

In the same year, Wong and colleagues in their placebo-controlled, double-blind, randomized crossover study enrolled 19 overweight/obese patients and postmenopausal women (BMI 25–35 kg/m^2^, aged 30–70 years) with blood hypertension [34]. They administered three doses of resveratrol (30, 90, or 270 mg once a day) or placebo for 4 weeks. The results of this study showed a linear relationship between the dose of resveratrol administered and its plasma concentrations. At the dose of 270 mg administered once a day, flow-mediated dilation (FMD) of the brachial artery increased in a dose-related manner [34]. In 2013, the same group of authors conducted another randomized placebo-controlled, cross-over study to evaluate the effects of resveratrol at the dose of 75 mg once daily to 28 healthy obese patients (BMI 30–45 kg/m^2^, aged 40–75 years) [35]. After 6 weeks of resveratrol administration, FMD increased by more than 23%. No adverse effects were reported [35].

In 2013, Poulsen and colleagues enrolled 24 male volunteers (aged 18 to 70 years) with obesity (BMI > 30 kg/m^2^) and no other diseases. Patients were divided into two groups: placebo (n = 12, age 31.9 ± 2.9 years) and resveratrol 500 mg three times a day (n = 12, age 44.7 ± 3.5 years) given for 4 weeks [36]. At the end of the study, the authors found no effects on glucose homeostasis, glycosylated hemoglobin (HbA1c), insulin sensitivity, lipid profile, blood pressure, total body fat mass, resting energy expenditure, glucose or lipid oxidation rate. Based on these results, they concluded that resveratrol has no effects in obese patients with moderate insulin-resistance [36]. Similar results were reported by van der Made and colleagues who conducted a randomized, placebo-controlled cross-over study on 45 overweight/obese patients (25 men, 20 women, BMI of 25–35 kg/m^2^, aged 45–70 years) with type 2 diabetes mellitus. Patients received either placebo or resveratrol 150 mg once a day for 4 weeks separated by a 4-week washout period. No change in metabolic parameters (fasting glucose, insulin, and lipid profile), inflammation markers, and endothelial function i.e., interleukin-6 (IL-6), tumor necrosis factor α (TNF α), soluble vascular adhesion molecule-1 (sVCAM-1), and soluble intracellular adhesion molecule-1 (sICAM-1) was observed. The authors also evaluated general health parameters, such as those assessing liver and kidney function, blood count, and coagulation parameters, and they found that alkaline phosphatase (ALP) increased significantly after administration of resveratrol. Resveratrol was well tolerated and no adverse reactions were reported [37]. 

The same dose of resveratrol (150 mg once daily) was used in a 2013 randomized, double-blind, placebo-controlled study. Ten obese patients (BMI 32 ± 1 kg/m^2^, aged 52 ± 2 years) were enrolled and received either placebo or resveratrol for 30 days after a 4-week wash-out period. All the patients enrolled underwent assessment of fasting and postprandial glucagon, glucagon-like peptide-1 (GLP-1), and glucose-dependent insulinotropic polypeptide (GIP) levels. The authors concluded that the polyphenol did not affect fasting and postprandial incretin hormone concentrations, but it suppressed postprandial secretion of glucagon [38].

Dash and colleagues, in their placebo-controlled crossover study, evaluated the effects of resveratrol on lipoprotein kinetics in eight overweight/obese subjects (BMI 31.1 ± 1.7 kg/m^2^, age 45.8 ± 3.1 years) with mild hypertriglyceridemia after 2 weeks of administration: 1000 mg per day the first week followed by 2000 mg per day the second [39]. The authors found no changes in insulin sensitivity and triglyceride concentration, but they observed a significant reduction in apoB-48 and apoB-100, which are lipoprotein particles produced by the intestine and liver, respectively [39].

In a 2014 randomized, placebo-controlled, double-blind study, 32 overweight/obese patients (BMI 25–34.9 kg/m^2^, mean age 73 ± 7 years) were divided into three groups; the first receiving placebo, the second a moderate dose of resveratrol (300 mg per day), and the third a high dose of resveratrol (1000 mg per day) [40]. All subjects underwent venous sampling to evaluate blood count and the metabolic profiles before and after placebo or resveratrol administration. After 90 days of treatment, fasting glucose and bilirubin levels were significantly lower in patients treated with resveratrol than in those receiving placebo, but not compared to the values at study enrollment. However, a decrease in hemoglobin levels and mean corpuscular volume was observed in the group of patients receiving 300 per day of resveratrol, whereas the patients given the highest dose had an increase in alkaline phosphatase and aspartate aminotransferase levels [40]. Therefore, these results highlighted the need for further studies to better understand the long-term safety of resveratrol supplementation.

Konings and colleagues, integrating their previous study [13], showed that the administration of 150 mg per day of resveratrol for 30 days to 11 obese but otherwise healthy patients decreased the size of subcutaneous adipocyte of the abdominal area and enhanced and improved adipogenesis, probably by modulating gene expression. Indeed, the authors found that resveratrol causes downregulation of some genes involved in intercellular junction, Wnt signaling, angiogenesis, G protein-coupled receptors, and Notch signaling. At the same time, resveratrol has upregulated pathways involved in cell cycle regulation and is associated with lysosome, phagosome, inflammation, and glucose transport [41].

More recently, in a randomized, placebo-controlled pilot clinical study, 28 obese patients (BMI between 30 and 40 kg/m^2^, age between 30 and 70 years) with metabolic syndrome were treated with resveratrol at the dose of 1000 mg twice a day for 30 days. All patients underwent an anthropometric evaluation, blood pressure monitoring, resting energy expenditure assessment, abdominal fat biopsy aspiration, 75-oral glucose tolerance test, euglycemic hyperinsulinemic clamp determination, and fecal microbiome analysis [42]. At the end of the observation period, resveratrol was well tolerated but the authors found no substantial effects on glucose homeostasis except for a lower 120 minutes glucose concentration in patients treated with resveratrol compared to the placebo group. A small Caucasian subgroup showed improvement in insulin sensitivity and glucose homeostasis after a 2-h oral glucose tolerance test [42]. The fecal microbiota was altered by the administration of resveratrol [42].

In 2018, the largest randomized, placebo-controlled clinical trial on the effects of resveratrol was conducted. The authors enrolled 112 overweight/obese patients with insulin-resistance (age ranging between 18 and 70 years; BMI ≥ 27 kg/m^2^, HOMA index ≥ 2), who received either placebo or 75 mg twice a day of resveratrol for 12 weeks. The percentage of liver fat content, OGTT with the evaluation of Matsuda’s insulin sensitivity index, HbA1c, lipid profile, liver function indexes, adiponectin, IL-6, cardiorespiratory fitness (VO_2max_), carotid intima-media thickness, and systolic and diastolic blood pressure were evaluated. No changes in cardiometabolic risk markers and liver fat content were found. Resveratrol was well tolerated and no adverse events were reported [43].

In recent months, the role of resveratrol was investigated in association with a physical activity and diet program [4]. Batista-Jorge and colleagues in their randomized, placebo-controlled study enrolled 25 obese patients (BMI ≥ 30 kg/m^2^, age range between 30–60 years) with metabolic syndrome who were randomly assigned to the placebo or resveratrol group (250 mg per day). All patients followed a physical activity program and a personalized diet [4]. After 3 months, BMI and waist circumference were significantly lower in both groups, however, only the resveratrol group showed an increase in HDL and a significant decrease of very-low-density lipoprotein (VLDL), total cholesterol, urea, creatinine, and albumin [4]. 

Very recently, a randomized, double-blind, placebo-controlled study evaluated the effects of resveratrol supplementation (150 mg per day for 6 months) in 41 overweight patients (BMI ranging between 27–35 kg/m^2^, age between 40–70 years). The authors found no improvement in insulin sensitivity but HbA1c decreased significantly [44]. No effect was recorded on intrahepatic lipid content, body composition, blood pressure, resting energy metabolism, and quality of life and/or sleep [44].

Table 1 summarizes the results of the main clinical trials on the effects of resveratrol on obesity.

## 5. Conclusions

Resveratrol is a polyphenolic compound with pleiotropic activities. Experimental data have shown cardioprotective, anticancer, antioxidant, anti-lipid, anti-aging, antibacterial, antiviral, and neuroprotective effects. Many preclinical studies have also reported that resveratrol improves glucose metabolism and homeostasis, thus suggesting its possible use as a therapeutic strategy in the fight against obesity. However, the data from human studies are still scarce and, at times, contradictory and inconclusive. This may also be due to the different dosages of resveratrol and the various lengths of treatment used in these clinical trials. The present review investigated the effects of resveratrol supplementation on obesity, considering the possible mechanisms of action of this polyphenol. Further studies will be needed to evaluate the long-term effects and on larger sample sizes to better understand the effects of resveratrol in-vivo and its mechanisms of action.

## Figures and Tables

**Figure 1 ijms-22-04362-f001:**
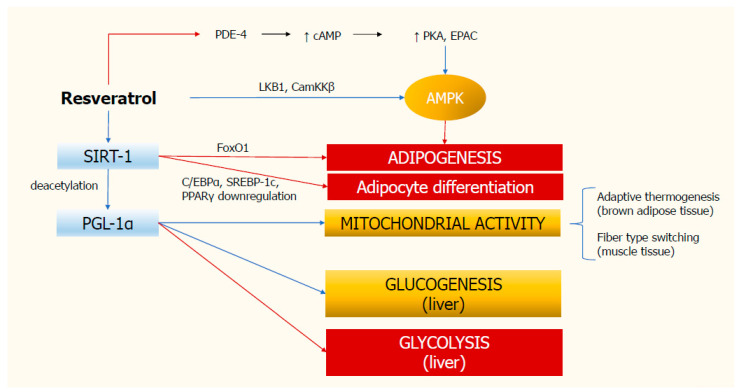
**Molecular mechanisms of resveratrol action.** Resveratrol stimulates SIRT-1 deacetylase, which in turn inhibits adipogenesis and adipocyte differentiation. SIRT-1 activates PGL-1α, thus triggering mitochondrial activity and glucogenesis, and inhibiting glycolysis in the liver. Resveratrol also increases the activity of AMPK, which hinders adipogenesis. Red rows indicate inhibition; blue rows indicate stimulation. AMPK, monophosphate-activated protein kinase; CamKKβ, calcium/calmodulin-dependent kinase kinase β; cAMP, cyclic adenosine monophosphate; C/EBPα, CCAAT-enhancer-binding protein; EPAC, exchange protein activated by cAMP; FoxO1, Forkhead box protein O1; PDE-4, type 4 phosphodiesterase; PKA, protein kinase A; PGC-1α, peroxisome proliferator-activated receptor γ coactivator-1α; PPARγ, proliferator-activated receptor-gamma; SIRT-1, deacetylating enzyme sirtuin-1; SREBP-1c, sterol regulatory element-binding protein 1c.

**Table 1 ijms-22-04362-t001:** Clinical trials on the effects of resveratrol on human obesity.

Authors	Type of Study	Patients	Resveratrol Dose	Duration of Resveratrol Administration	Conclusions
Timmers et al., 2011	Randomized, placebo-controlled, double-blind cross-over	11 obese otherwise healthy men	150 mg once daily	30 days	Reduction of glucose and insulin serum levels Reduction of HOMA index and TGL Improved sleeping and resting metabolic rate, blood pressure, and liver function Improved mitochondrial function Upregulation of gene pathway related to mitochondrial oxidative phosphorylation and downregulation of pathways related to inflammation No effect on body weight No adverse reactions
Crandall et al., 2011	Randomized, open-label uncontrolled	10 overweight/obese adults with moderate insulin-resistance	1.0 g once daily 1.5 g once daily 2.0 g once daily	4 weeks	Significant reduction of post-meal peak of glucose No change of fasting glucose, insulin secretion and HOMA index, No effects on body weight, blood pressure, fasting lipid profile, reactive C-protein, and adiponectin
Wong et al., 2011	Randomized, placebo-controlled, double-blind, cross-over	19 overweight/obese adults with hypertension	30 mg once daily 90 mg once daily 270 mg once daily	4 weeks	Increased FMD after administration of resveratrol 270 mg FMD increase in dose-related manner
Wong et al., 2013	Randomized, placebo-controlled, double-blind, cross-over	28 obese otherwise healthy adults	75 mg once daily	6 weeks	Increased FMD by 23%
Poulsen et al., 2013	Investigator-initiated, randomized, place-controlled	24 male volunteers with obesity but otherwise healthy	500 mg three times a day	4 weeks	No effect on HbA1c and HOMA index No effect on total cholesterol, HDL, LDL, and TGL No change in resting energy expenditure, body weight, and total fat mass No effects on blood pressure
Van der Made et al., 2015	Randomized, placebo-controlled, cross-over	45 overweight/obese patients with DM2	150 mg once daily	4 weeks	No effect on metabolic risk parameters No changes of plasma markers of inflammation and endothelial function No adverse reactions
Knope et al., 2013	Randomized, placebo-controlled, double-blind, cross-over	10 obese men	150 mg once daily	30 days	No effect on fasting and post-prandial incretin hormone concentrations Suppressed postprandial glucagon secretion
Dash et al., 2013	Randomized, placebo-controlled, double-blind, cross-over	8 overweight/obese subjects with mild hypertriglyceridemia	1000 mg once daily for a week followed by 2000 mg once daily for another week	2 weeks	No changes in insulin sensitivity and TGL concentration Reduction of apoB-48 and apoB-100
Anton et al., 2014	Randomized, placebo-controlled, double-blind	32 overweight/obese adults	Three groups: placebo, moderate resveratrol dose 300 mg once daily, and high resveratrol dose 1000 mg once daily	90 days	Improved glucose and bilirubin vs. placebo Decreased hemoglobin and mean corpuscular volume in moderate dose group Increased AST and alkaline phosphatase in high-dose group
Konings et al., 2014	Randomized, placebo-controlled, double-blind, cross-over	11 obese otherwise healthy men	150 mg once daily	30 days	Reduction of adipocytes size, enhanced and improved adipogenesis
Walker et al., 2018	Randomized, placebo-controlled	28 obese men with metabolic syndrome	1000 mg twice a day	30 days	No substantial changes of glucose homeostasis
Kantartzis et al., 2018	Randomized, placebo-controlled	112 overweight/obese and insulin-resistant patients	75 mg twice a day	12 weeks	No effects on cardiometabolic risk markers and on liver fat content
Batista-Jorge et al., 2020	Randomized, placebo-controlled	25 obese adults with metabolic syndrome	250 mg once daily + physical activity program + diet	3 months	Reduction of VLDL, HDL, total cholesterol, urea, creatinine, and albumin
de Ligt et al., 2020	Randomized, placebo-controlled, double-blind	41 overweight/obese adults	150 mg once daily	6 months	No effects on insulin sensitivity Reduction of HbA1c

HOMA = homeostasis model assessment; TGL = triglycerides; FMD = flow mediated dilatation of the brachial artery; HbA1c = glycosylated hemoglobin; HDL = high density lipoprotein; LDL = low density lipoprotein; DM2 = diabetes mellitus type 2; AST = aspartate aminotrasferase; VLDL: very-low density lipoprotein.

## Data Availability

Data sharing not applicable.

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
