# Peer review of "The Role of Resveratrol Administration in Human Obesity"

_ijms, 2021, doi:10.3390/ijms22094362_

Round 1

Reviewer 1 Report

This narrative review summarizes recent literature investigating the effects of resveratrol on numerous biochemical factors relevant to obesity and metabolic syndrome. Reviews examining experimental evidence related to bioactive compounds in foods are highly relevant to public health due to the fact that the effects of these compounds are often inaccurately reported in the media and can lead the public to erroneous dietary practices in an effort to increase their intake of compounds with purported health benefits. Resveratrol is particularly troublesome in this regard because the public can easily increase intake of such products as red wine to an excessive amount in a misguided attempt to consume resveratrol. 

This review is logically organized and very well written. The authors have used solid judgment in their inclusion of a range of studies that are useful in providing insight the mechanisms of action of resveratrol and results of  human trials that address the topic from a clinical perspective. The figure and table are also well presented and useful. 

A bit more perspective on inhibition of adipogenesis and adipocyte differentiation would enhance the paper. While the connection between adipogenesis and obesity is clear and merits extensive treatment in this review, the authors could improve the discussion of this specific topic by explicitly considering the literature indicating that promoting adipocyte differentiation can be beneficial in some contexts. For example, during periods of excessive energy intake, adipocyte function should increase (particularly in subcutaneous depots) to enable safe storage of excess substrate in adipocytes as neutral lipids, thereby decreasing the possibility of ectopic lipid deposition in tissues such as the liver. There is a body of work highly relevant to this review that has examined botanically derived supplements for their potency in upregulating several factors promoting adipogenesis and neutral lipid deposition in different fat depots in rodents and in cell culture studies. (Richard, AJ, et al. Artemisia scoparia enhances adipocyte development and endocrine function in vitro and enhances insulin action in vivo. 2014, PLoS One; Boudreau, A. Mechanisms of Artemisia scoparia's Anti-Inflammatory Activity in Cultured Adipocytes, Macrophages, and Pancreatic β-Cells. Obesity, 2020). These studies provided evidence of a clear linkage between enhanced adipogenesis and metabolically advantageous alterations in glucose homeostasis and systemic insulin sensitivity. 

The text of the manuscript is very readable and well written. One minor stylistic change is recommended:

Line 86 - change to "in animal models"

Author Response

Answer to comment 1: Thank you very much for your comments and your appreciation. We agree with you regarding the beneficial effects of adipocyte differentiation; consequently, we further discussed this aspect and added the studies you suggested. Please see the paragraph “Resveratrol and obesity: possible mechanisms of action” (lines 174-186, reference numbers 31 and 32).

Answer to comment 2: We corrected the expression in line 86, as you requested. You can find our changes highlighted in yellow.

Reviewer 2 Report

The manuscript describes a number of studies focusing on the role of resveratrol in human obesity, including studies with placebo controls, and those taking into consideration age, gender, physical activity, and following metabolic changes in participants.

Few typos need to be fixed before the review is published. 

The Figure needs to be incorporated in the text, right now it is in the "non-published material" section.

Author Response

Comment 1: Few typos need to be fixed before the review is published.

Answer to comment 1: Thank you very much for your comments and your appreciation. We revised the text to correct typos.

Comment 2: The Figure needs to be incorporated in the text, right now it is in the "non-published material" section.

Answer to comment 2: The Table was incorporated in the text after the paragraph “Discussion” and before the paragraph  “Conclusion”. You can find our changes highlighted in yellow.

This manuscript is a resubmission of an earlier submission. The following is a list of the peer review reports and author responses from that submission.